# Antimicrobial Activity of an Extract of *Hermetia illucens* Larvae Immunized with *Lactobacillus casei* against *Salmonella* Species

**DOI:** 10.3390/insects11100704

**Published:** 2020-10-15

**Authors:** Kyu-Shik Lee, Eun-Young Yun, Tae-Won Goo

**Affiliations:** 1Department of Pharmacology, College of Medicine, Dongguk University, Gyeongju 38766, Korea; there1@dongguk.ac.kr; 2Department of Integrative Bio-Industrial Engineering, Sejong University, Seoul 05006, Korea; yuney@sejong.ac.kr; 3Department of Biochemistry, College of Medicine, Dongguk University, Gyeongju 38766, Korea

**Keywords:** antimicrobial activity, natural antibiotics, antimicrobial peptide, *Hermetia illucens*, *Salmonella*

## Abstract

**Simple Summary:**

In this investigation, we found antimicrobial activity of the extract of *Lactobacillus casei*–infected *Hermetia illucens* larvae (HIL) against *Salmonella* species. The result demonstrates that the extract is a powerful natural antibiotic and preservative.

**Abstract:**

The expressions of antimicrobial peptides (AMPs) in the larvae of the black soldier fly, *Hermetia illucens*, were significantly increased by pathogen or stimulant induced innate immunity activation. We immunized *H. illucens* fifth instar larvae with five different *Lactobacillus* species, that is, *Lactobacillus acidophilus*, *L. brevis*, *L. casei*, *L. fermentum*, or *L. delbrueckii*, to induce the mass production of AMPs and selected optimal immune inducers. Antimicrobial activities in hemolymph and *H. illucens* larvae (HIL) extract were evaluated against three *salmonella* species (*Salmonella pullorum*, *Salmonella typhimurium*, and *Salmonella enteritidis*). Highest antimicrobial activity was shown by the hemolymph of HIL immunized by *L. casei* and its activity was closely linked with the inductions of cecropin 1 (HiCec1) and defensin 1 (HiDef1) gene expressions. Furthermore, antimicrobial activity in hemolymph was stable to heat and pH and the growth of three *Salmonella* species were dramatically suppressed by HIL hemolymph and extract after immunization with *L. casei*. The minimal inhibitory concentration (MICs) of *L. casei*-immunized HIL extract against *Staphylococcus aureus*, *Escherichia coli*, and *Salmonella* species ranged from 100~200 µg/100 µL and no cytotoxicity to CaCo-2 and L929 cells were observed in the concentration range 100~40,000 µg/100 µL. Taken together, the present investigation demonstrates that *L. casei*-immunized HIL extract is a powerful natural antibiotic and preservative that can prevent contamination by *Salmonella* species.

## 1. Introduction

Antibiotics are used to treat infectious diseases and prevent infections in humans and animals and to promote animal growth and thus have played leading roles in the intensive farming of livestock. However, abuse of in-feed antibiotics has caused many problems, such as the appearance of antibiotic-resistant bacteria, antibiotic residues in livestock products, weakening of disease resistance in livestock, and ecosystem pollution [1,2,3]. To address these problems, the European Union prohibited the use of veterinary antibiotics for growth promotion from 2006, whereas in Korea, after a phased reduction from 2005, their usage was banned in July 2011. Nonetheless, antibiotic offtake for treatment purposes is expected to increase due to livestock disease outbreaks caused by the prohibition of veterinary antibiotics in feed. Therefore, novel natural antibiotics are needed to overcome antibiotic-resistant bacteria [4,5].

Insects, such as *Hermetia illucens*, have evolved innate immune systems that produce potent antimicrobial peptides (AMPs) to protect them from pathogen invasion, and these AMPs are viewed as strong natural antibiotic candidates [6,7]. The insect innate immune system is characterized by cellular and humoral immunity. Cellular immunity involves the phagocytosis of bacteria, fungi, and protozoa, and nodule formation and encapsulation, whereas humoral immunity involves the secretions of proteins and peptides produced in fat and blood cells to hemolymph in response to infection [7,8,9,10]. AMPs secreted by the humoral immune response are classified by amino acid sequence and structure as cecropins, defensins, proline-rich peptides, glycine-rich peptides, and lysozymes and are found in various insects including Coleoptera, Diptera, Hymenoptera, and Lepidoptera [11,12,13,14,15,16]. Melittin is well-known AMP contained in bee venom and its antimicrobial activity was fully elucidated in methicillin-resistant *Staphylococcus aureus* and Gram-positive and Gram-negative bacteria [17,18,19]. *H. illucens* survives in organic wastes containing bacterial, fungi, and viruses, and thus produces potent antimicrobials to protect itself against from these entities. Furthermore, *H. illucens* grows rapidly and its feed costs are low, which makes it suitable for the mass production of natural antibiotics. However, *H. illucens* expresses/secretes AMPs in quantity only after innate immune system activation by a pathogen [20,21]. Choi et al. previously demonstrated the antibiotic activity of purified peptide extract in hemolymph isolated from *H. illucens* larvae (HIL) immunized by *Lactobacillus casei* against *Klebsiella pneumoniae* (ATCC 13883) and *Shigella dysenteriae* (ATCC 9750) [22]. Here, to select an optimal AMPs inducer, we evaluated antimicrobial activity of HIL immunized with five *Lactobacillus* species and assessed the bactericidal activities of an extract of HIL immunized with a selected *Lactobacillus* species to determine the potential use of the extract as a natural antibiotic or preservative.

## 2. Materials and Methods

### 2.1. H. illucens Larvae and Bacteria Strains

HIL were gifted by the Department of Agricultural Biology at the National Institute of Agricultural Sciences of the Rural Development Administration (Wanju, Korea). Animals were maintained at room temperature (RT, 26 ± 1 °C) and 60% RH. Five *Lactobasillus* species, that is, *L. acidophilus* (KCCM 12419), *L. brevis* (KCCM 10553), *L. casei* (KCCM 12413), *L. fermentum* (KCCM 11441), and *L. delbrueckii* (KCCM 35463), were obtained from the Korean Collection for Type Culture (Wanju, Korea) and used to activate the innate immune system of *H. illucens*. The antimicrobial activity of HIL hemolymph was assessed using the Gram-positive bacterium *S. aureus* (KCCM 40881) and the Gram-negative bacteria *Escherichia coli* (KCCM 11234) and *Salmonella pullorum* (KVCC-BA0702509), *Salmonella typhimurium* (KCCM 40406), and *Salmonella enteritidis* (KCCM 12021), which were also obtained from the National Institute of Animal Science of the Rural Development Administration (Wanju, Korea).

### 2.2. Immunization of HIL

Using a fine needle (diameter 0.35 mm, length 40 mm), 10^1^ to 10^9^ cfu/mL of *Lactobacillus* species were inoculated into *H. illucens* fifth instar larvae by abdominal puncture and then the larvae were held on RT for 2, 4, 8, 16, 24, 48, or 72 h with starvation. Hemolymph was isolated and assessed for antimicrobial activity and AMP mRNA expressions.

### 2.3. Preparation of HIL Extracts

Immunized HIL were dried for 45 min using a by microwave and ground to a powder, which was suspended in 20% acetic acid solution, boiled for 30 min, and centrifuged at 4500 rpm for 30 min at 4 °C. Supernatant was collected, dried for 9 h in vacuum-spin drier, and resuspended in sterilized distilled-water as HIL extract.

### 2.4. Evaluation of the Antimicrobial Activities in HIL Hemolymph and Extract

The antimicrobial activities of HIL extracts were assessed using a radial diffusion assay (RDA) using Gram-positive *Staphylococcus aureus* and Gram-negative *Escherichia coli*, *S. pullorum*, *S. typhimurium*, and *S. enteritidis*. To perform RDA, autoclaved underlay gel [9 mM sodium phosphate, 1 mM sodium citrate, pH 7.4, 1% low electroendosmosis agar, and 0.03% tryptic soy broth (TSB)] was mixed with each bacterium and hardened in a 100 mm square plate. Then, we prepared 3.5 mm-diameter of well onto underlay gel and 10 µL of hemolymph or extract of HIL was injected into the well. The underlay gel was held at 37 °C for 3 h and then covered with a sterilized overlay gel (6% TSB and 1% low electroendosmosis agar). Gel plates were incubated for 18 h at 37 °C, and antimicrobial activities were assessed by measuring the widths of clear zones.

### 2.5. Determination of Minimum Inhibitory Concentrations (MICs) of HIL Extracts

Firstly, we determined cfu/mL of each bacterial by serial tittering and then measured absorbance of each cfu/mL at 600 nm. Then, *S. aureus* and *E. coli* were used to determine the MICs of extracts obtained after immunization with *L. casei*. Determination of MICs was also used for evaluation of antimicrobial activity against Salmonella species. Bacteria, such as *S. aureus*, *E. coli*, *S. pullorum*, *S. typhimurium*, and *S. enteritidis*, were inoculated into liquid medium, respectively, and grown at 37 °C for 18 h in a shaking incubator (shaking speed: 200 rpm), and then further cultured to 4 × 10^6^ cfu/mL over 2.5 h under the same conditions. A total of 90 µL (1 × 10^6^ cfu/mL) of bacterial culture mixtures were transferred to a 96-well plate and 10 µL of serially diluted extracts were then added to wells and incubated at 37 °C for 18 h. Absorbances were measured at 600 nm to determine MICs. Melittin was purchased from Sigma-Aldrich; Merck KGaA (Darmstadt, Germany) was used as control.

### 2.6. Analysis of AMPs Transcription

*Lactobacillus*-infected HIL were held at RT for defined times and then total RNA was extracted. First-strand cDNA was synthesized using a high capacity reverse transcription kit (Applied Biosystems, Waltham, MA, USA), and the cDNA obtained was assessed for the gene expressions of cecropin 1 (HiCec1) and defensin 1 (HiDef1). Primers for HiCec1 and HiDef1 were designed using the Primer 3 program (http://simgene.com/Primer3; accessed on 13 June, 2016) and gene expressions were normalized versus the *Drosophila melanogaster* actin 5C (DmAct5C) gene. Primer sequences are provided in Table 1.

### 2.7. Cytotoxicity Analysis by MTT Assay

We investigated the cytotoxicity of *Lactobacillus*-immunized HIL extract using CaCo-2 human intestinal cells and L929 mouse fibroblasts. Cells were seeded in a 96-well plates and allowed to attached for 24 h in a 5% CO_2_ atmosphere at 37 °C. Then, culture medium was removed and then cells were treated with 180 µL of various concentrations of *Lactobacillus*-immunized HIL extract (0~5000 µg/100 µL) or melittin (0~32 µg/µL) for 24 h. Cells were then administered 20 µL of 5 mg/mL MTT reagent and left for 4 h in the dark. Media were then carefully discarded and 200 µL of DMSO was added to each well to dissolve the formazan. Absorbances were measured at 540 nm using an ELISA reader.

### 2.8. Statistical Analysis

The analysis was performed using the Student’s *t*-test and one-way analysis of variance. Experiments were independently performed at least three times, and results are presented as means ± standard deviations (SDs). *p*-value of less than 0.05 was regarded as statistically different.

## 3. Results

### 3.1. Optimization of Lactobacillus Species for the Mass Production of AMPs in H. illucens

We used five types of *Lactobacillus* to activate the innate immune system of *H. illucens*. RDA results showed normal HIL hemolymph did not exhibited antimicrobial activity against *E. coli* or *S. aureus* (Figure 1). Furthermore, we did not observed death of any HIL infected by *Lactobacillus* species (data not shown). In contrast, although the induction of antimicrobial activity was depended on *Lactobacillus* species type, increased antimicrobial activities to *E. coli* and *S. aureus* were observed in the hemolymph of *Lactobacillus*-immunized HIL (Figure 1). Highest antimicrobial activity against *E. coli* and *S. aureus* were observed for the hemolymph of *L. casei*-infected larvae, and notably, this was greater than that of *Enterococcus faecalis* (KACC 11859) or *Serratia marcescens* (KACC 11961) challenged larvae (Figure 1) [23]. Therefore, *L. casei* was chosen as an activator of the innate immune system of *H. illucens*.

### 3.2. Determination of Optimal Conditions for the Mass Production of AMPs

The results obtained showed that antimicrobial activity in hemolymph was increased by *L. casei* infection in a concentration-dependent manner (Figure 2A). We also evaluated changes in antimicrobial activity over the time after *L. casei* infection. As shown in Figure 2B, highest activity was observed at 24 h after infection. Furthermore, the gene expressions of HiCec1 and HiDef1 (major AMPs produced by *H. illucens*) peaked at 24 h (Figure 2C). This result showed that the antimicrobial activity of *L. casei*-infected HIL hemolymph was closely linked with the induced gene expressions of HiCec1 and HiDef1.

### 3.3. Analysis of the Antimicrobial Activities of HIL Hemolymph against Salmonella Species

We investigated the antimicrobial activities of *L. casei*-immunized HIL hemolymph against three *Salmonella* species using RDA. *Salmonella* species are pathogens parasitic on gut and major causes of food poisoning. In fact, 75% of food poisoning cases are due to *Salmonella* species, and reductions in livestock productivity are closely associated with *Salmonella* contamination [24,25]. We found that the antimicrobial activities of HIL hemolymph against *S. enteritidis*, *S. typhimurium*, and *S. pullorum* were dramatically enhanced by *L. casei*-immunization and that their activities were greater than that of 1 µg of melittin (Figure 3A). Furthermore, the antimicrobial activity of *L. casei*-immunized HIL hemolymph was maintained after heating (60, 70, 80, or 90 °C) for 24 h and over the pH range 2~10 (Figure 3B,C). We also found that antimicrobial activity of hemolymph was lowest at pH 7.0 (Figure 3C). These results suggest that AMPs produced by *L. casei*-immunized HIL effectively suppress the growth of *Salmonella species* and is stable to heat and a wide pH range.

### 3.4. Determination of the MICs of L. casei-Immunized HIL Extract

We investigated whether *L. casei*-immunized HIL extract has the potential to be used as a natural antibiotic or feed additive with antimicrobial activity. Antimicrobial activities were evaluated by determining MICs against *E. coli* and *S. aureus*. The MIC range of melittin to *E. coli* and *S. aureus* was 7~10 µg/100 µL and the *L. casei*-immunized HIL extract had an MIC range of 100~200 µg/100 µL (Figure 4A,B). The result reveals that both bacteria, Gram-negative *E. coli* and Gram-positive *S. aureus*, show same sensitivity to antimicrobial substances in the extract. Although MICs of the extract were greater than that of melittin, the manufacturing cost of *L. casei*-immunized HIL extract is much lower than that of melittin, and thus, the extract is more cost-effective.

### 3.5. Analysis of the Antimicrobial Activities of L. casei-Immunized HIL Extract against Salmonella Species

We assessed the antimicrobial activities of *L. casei*-immunized HIL extract against *S. enteritidis*, *S. typhimurium*, and *S. pullorum* by determining MICs. The results obtained showed that th MICs of melittin to *Salmonella* species were 10 µg/100 µL and at a concentration of >200 µg/100 µL the extract completely blocked the growth of all *Salmonella* species (MICs range 100~200 µg/100 µL) (Figure 5A,B).

### 3.6. Analysis of the Cytotoxicities of L. casei-Immunized HIL Extracts to Mammalian Cells

The cytotoxic effects of antimicrobial in human and animals are key obstacles to clinical, livestock, and feed applications. Therefore, we examined the cytotoxic activities of melittin and of *L. casei*-immunized HIL extract in CaCo-2 and L929 cells (human intestinal and cell mouse fibroblast cell lines, respectively). No cytotoxic effect was observed on against CaCo-2 and L929 cells at extract concentrations from 100 to 4000 µg/100 µL or from 100 to 2000 µg/100 µL, respectively (Figure 6A). In contrast, although cell survival was not reduced by melittin at 1 to 4 µg/100 µL, it was found to have potent cytotoxic effects at > 8 µg/100 µL (Figure 6B).

## 4. Discussion

Synthetic antibiotics and antimicrobials have contributed to public health and promoted the growth of livestock. However, overuse and abuse of antibiotics and antimicrobial drugs are now known to be important causes of drug-resistant bacteria, which threaten public and livestock health. Natural antibiotics, especially AMPs, are considered as promising novel candidates for overcoming drug-resistance [26]. Several investigations have demonstrated insects manufacture AMPs and that these acts as potent natural antibiotics [7,11,12,27]. One investigation showed that a HIL extract prevented growth of plant pathogens and suggested its use as a natural antimicrobial [28]. Furthermore, *Lactobacillus* species activated innate immune response in insects and contributed to reduction of susceptibility in pathogen infected-insects [29,30,31,32]. In this investigation, we also found HIL extract immunized with *L. casei* exhibited antimicrobial activity against three *Salmonella* species (Figure 4B), which demonstrates that this extract offers a potential means of preventing and treating plant pathogen infections and protecting against *Salmonella* contamination.

In general, insect AMPs are induced when the innate immune system is activated and are present at much lower levels under normal conditions. In this investigation, we used five *Lactobacillus* species to activate the innate immunity system. As shown Figure 1, highest antimicrobial activities against Gram-positive *S. aureus* and Gram-negative *E. coli* were observed in the hemolymph of *L. casei*-infected *H. illucens*, and these activities were higher than those observed for the hemolymphs of HIL challenged by *E. faecalis* or *S. marcescens* (Figure 1). *E. faecalis* and *S. marcescens* are well known human and animal pathogens [33,34]. Although these bacteria also effectively activated the innate immune system of *H. illucens*, they pose risks to human and animal health. In contrast, *L. casei* is a beneficial bacterium in human and its use induced significant increases in HiCec1 and HiDef1 levels and superior antimicrobial activity in the hemolymph and the extract of HIL (Figure 2 and Figure 4). Furthermore, we found that the antimicrobial activity of hemolymph was stable for 24 h over the temperature range (60–90 °C) and the pH range (pH 2–11) (Figure 3B,C). In addition, the extract of HIL immunized with *L. casei* did not adversely affect the survival of L929 or CaCo-2 cells at concentrations up to 2000 µg/100 µL or 4000 µg/100 µL, respectively (Figure 6A). These results demonstrate that *L. casei*-immunized HIL offer a safe antibiotic candidate for the treatment of bacterial infections in human and livestock and for food preservation.

Melittin is major component of bee venom and has been reported to be a powerful antimicrobial against human and animal pathogens and against methicillin-resistant *S. aureus* [17,35,36]. However, the production cost of melittin and purified bee venom is very high and their applications in the medical and food industries are limited by cytotoxic effects on normal cells and allergic responses (Figure 6B) [37]. In the present study, cytotoxic concentration of melittin to Caco-2 and L929 were dramatically lower than MIC to *E. coli* and *S. aureus* (Figure 6B). In contrast, *L. casei*-immunized HIL extract could be produced cheaply and this extract proved to be relatively non-toxic to normal animal cells (Figure 6A). In fact, the MICs of *L. casei*-immunized HIL extract to *E. coli* and *S. aureus* were 20- to 50-fold lower than those to normal animal cells (Figure 4A and Figure 6A). Consequently, the present investigation shows that *L. casei*-immunized HIL extract provides a potential cost-effective means of producing a natural antibiotic with applications in medicine and livestock. However, the allergic effects of this extract have yet to be investigated.

To prevent *Salmonella* contamination and growth, synthetic preservatives, such as ethylparaben, methylparaben, benzoice acid, sorbic acid, and propanoic acid, are added to foodstuffs, including meat products [38]. Although synthetic preservatives are effective and economical, many have reported harmful effects [39], and thus, there is a considerable demand for a safe and economical natural preservative. We report the hemolymph and extract of *L. casei*-immunized HIL exhibit powerful antimicrobial activities against three *Salmonella* species (*S. enteritidis*, *S. typhimurium* and *S. pullorum*) as determined by RDA and MIC assays (Figure 3 and Figure 5). Furthermore, *L. casei*-immunized HIL extract was non-toxic to normal animal cells at concentrations up to 4000 µg/100 µL (Figure 6A). Our results demonstrate that *L. casei*-immunized HIL extract should be considered a potential natural preservative for the prevention of food poisoning caused by *Salmonella* species and a functional feed additive that promotes livestock growth.

## 5. Conclusions

Summarizing, we show that *Lactobacillus* can be used as an AMP inducer in HIL and that the extract of *L. casei*-immunized HIL prevents the growth of *Salmonella* species without inducing cytotoxicity in normal animal cells. Furthermore, the antimicrobial activity of *L. casei*-immunized HIL extract remained stable over the temperature (60–90 °C) for 24 h and the pH range (pH 2–11), respectively. In conclusion, the present investigation reveals that *L. casei*-immunized HIL extract is a potent natural antibiotic and preservative with potential applications the medical, food, and livestock industries.

## Figures and Tables

**Figure 1 insects-11-00704-f001:**
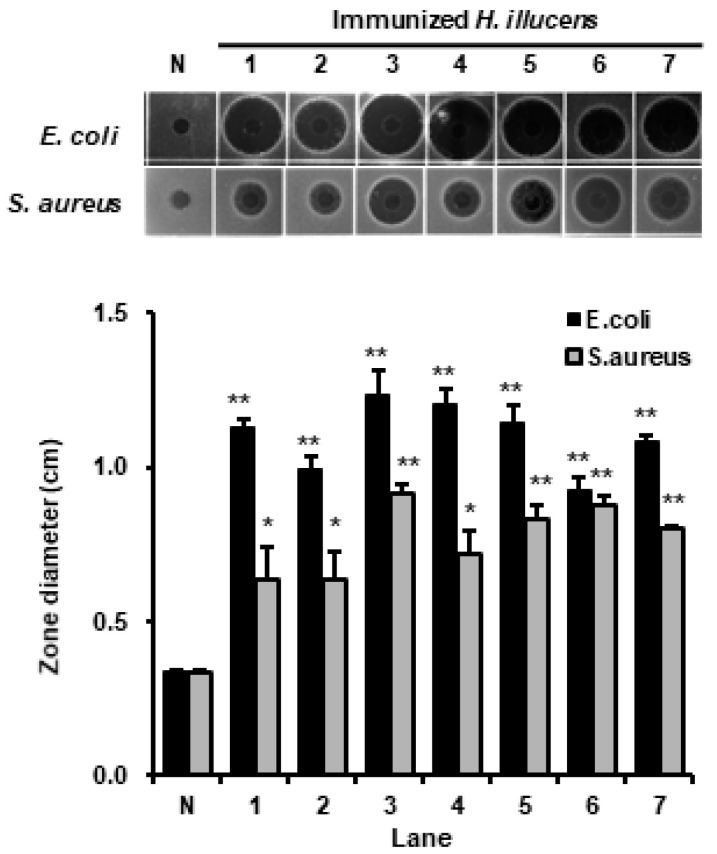
Comparison of antimicrobial activities to *E. coli* and *S. aureus* in the hemolymph of HIL immunized with *E. faecalis*, *S. marcescens*, *L. acidophilus*, *L. brevis*, *L. casei*, *L. fermentum*, or *L. delbrueckii*. N indicates non-immunized hemolymph of *H. illucens.* Lane 1, hemolymph of *S. marsescen-*immunized HIL; lane 2, hemolymph of *E. faecalis-*immunized HIL; lane 3, hemolymph of *L. casei-*immunized HIL; lane 4, hemolymph of *L. acidophilus-*immunized HIL; lane 5, hemolymph of *L. brevis-*immunized HIL; lane 6, hemolymph of *L. fermentum-*immunized HIL; lane 7, hemolymph of *L. delbrueckii-*immunized HIL. The values of column graphs are presented as the mean ± S.D. * and ** indicate *p* < 0.05 and *p* < 0.01 vs. normal control (N), respectively.

**Figure 2 insects-11-00704-f002:**
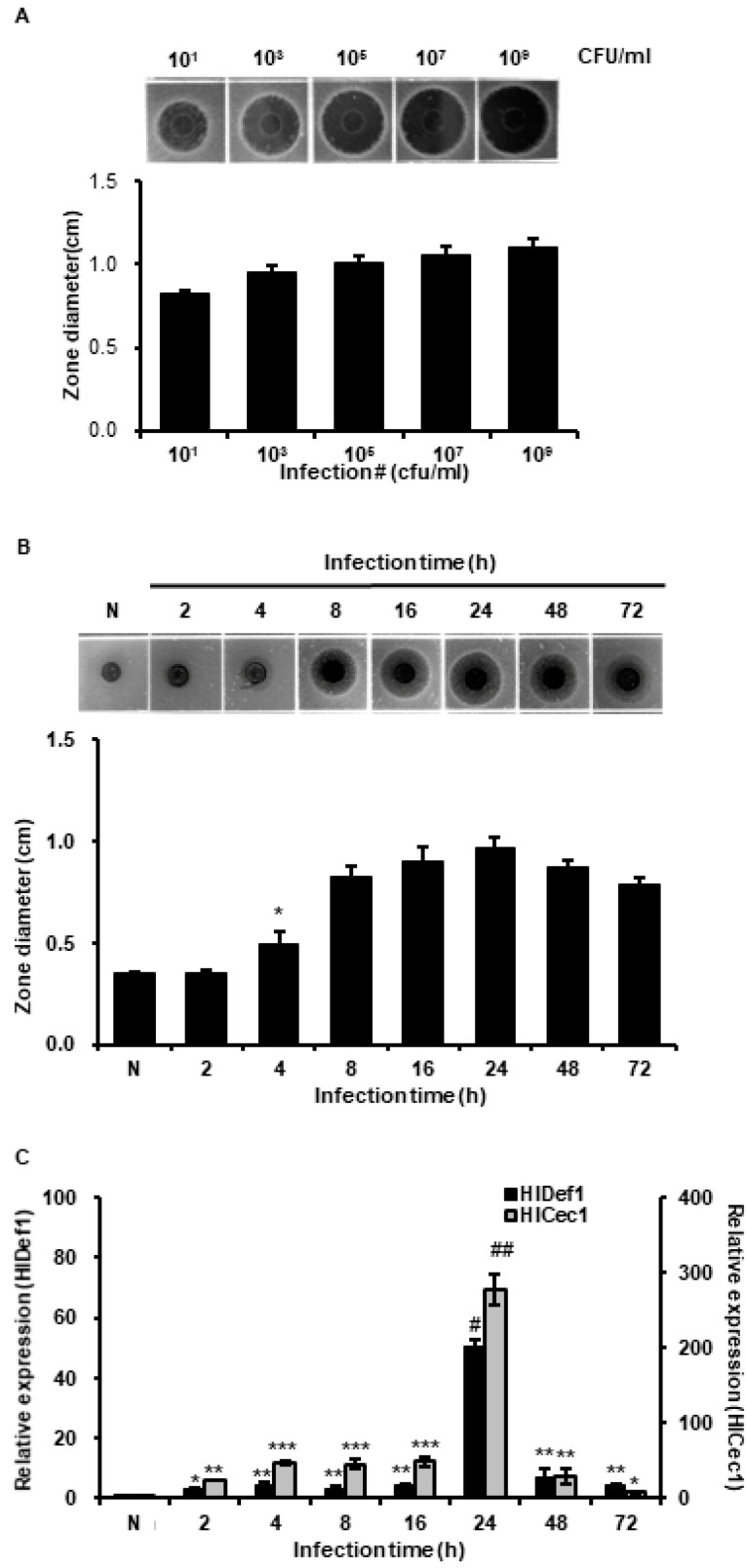
Determination of optimal conditions to produce *L. casei*-immunized *H. illucens* larvae (HIL) with highest antimicrobial activity. (**A**) Evaluation of antimicrobial activities at different *L. casei* concentrations. We infected HIL with 10^1^ to 10^9^ cfu/mL of *L. casei* to determine the optimal concentration for the mass production of AMPs. The activity in *L. casei*-immunized HIL hemolymph increased in a concentration-dependent manner. The values of column graphs are presented as the mean ± S.D. (**B**) Evaluation of antimicrobial activities at different incubation times after *L. casei* infection. Highest hemolymph activity of *L. casei*-immunized HIL occurred at 24 h. (**C**) Assessment of HiCec1 and HiDef1 mRNA expressions. HiCec1 and HiDef1 gene expressions in *L. casei*-immunized HIL hemolymph were evaluated at 2, 4, 8, 16, 24, 48 and 72 h after infection. (**A**,**B**) Radial diffusion assay (RDA) was performed with *E. coli*. (**B**,**C**) The values of column graphs are presented as the mean ± S.D. *, **, ***, # and ## indicate *p* < 0.05, *p* < 0.01, *p* < 0.005, *p* < 0.001 and *p* < 0.00001 vs. normal control (N), respectively.

**Figure 3 insects-11-00704-f003:**
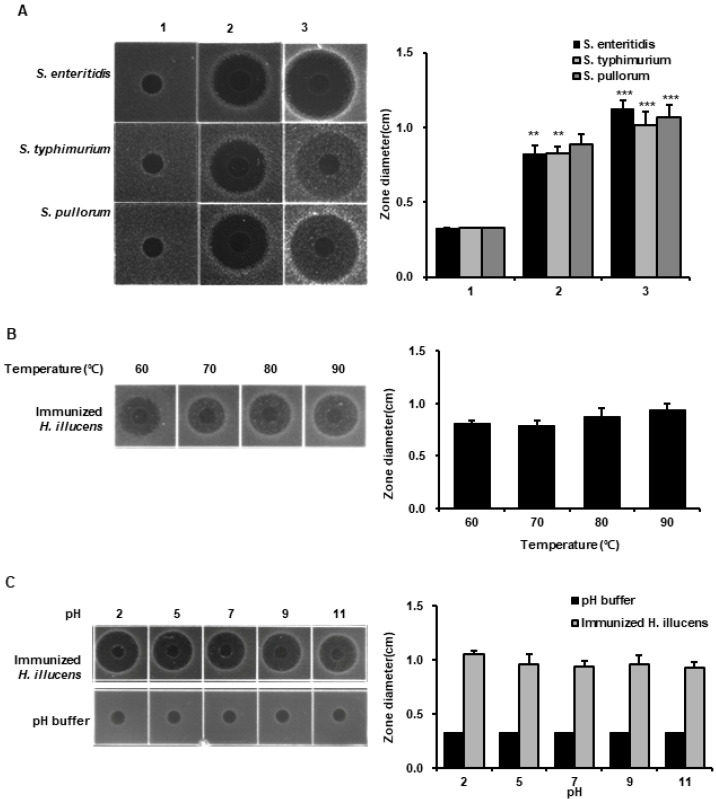
Evaluations of antimicrobial activities to *S. enteritidis*, *S. typhimurium* and *S. pullorum* and of the heat and pH stabilities of *L. casei*-immunized HIL hemolymph as determined by RDA. (**A**) Antimicrobial activities of *L. casei*-immunized HIL hemolymph against *S. enteritidis*, *S. typhimurium*, and *S. pullorum*. Lane 1, non-immunized HIL hemolymph; lane 2, melittin (1 µg); lane 3, *L. casei*-immunized HIL hemolymph. The values of column graphs are presented as the mean ± S.D. ** and *** indicate *p* < 0.01 and *p* < 0.005 vs. lane 1, respectively. (**B**) Thermal stabilities of the antimicrobial activities of hemolymphs. Hemolymphs were incubated at 60, 70, 80, or 90 °C for 24 h. (**C**) pH stabilities of antimicrobial activity of *L. casei*-immunized HIL hemolymph. A total of 2 µL of Hemolymphs was mixed in 8 µL of pH 2, 5, 7, 9, or 11 buffer for these evaluations. (**B**,**C**) *E.coli* was used as a target bacterium to confirm antimicrobial activity of *L. casei*-immunized HIL hemolymph.

**Figure 4 insects-11-00704-f004:**
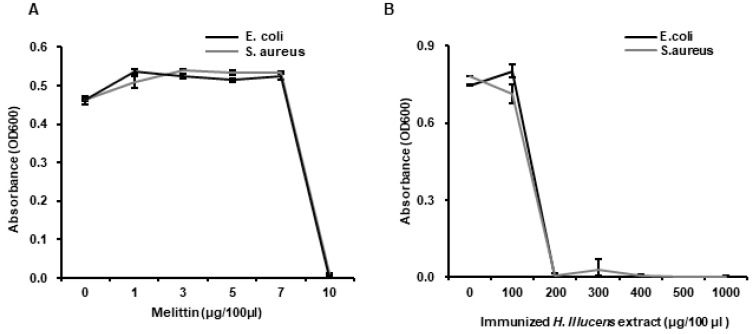
Determination of the minimal inhibitory concentrations (MICs) of melittin (**A**) and *L. casei*-immunized HIL extract (**B**). Various concentrations of 10 µL melittin or 10 µL *L. casei*-immunized HIL extract (10 µL) were mixed with 90 µL of liquid media containing 1 × 10^6^ cfu/mL of *E. coli* or *S. aureus* and incubated at 37 °C for 24 h.

**Figure 5 insects-11-00704-f005:**
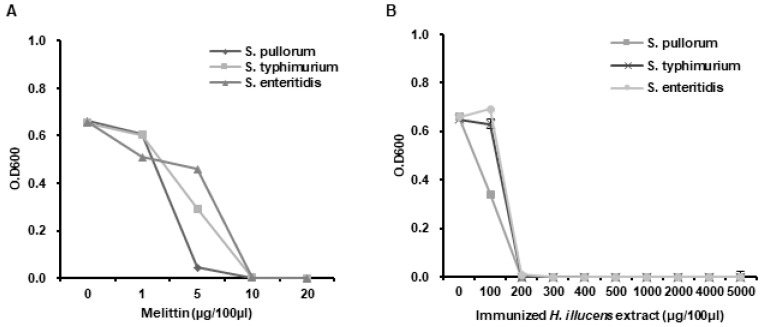
Antimicrobial activities of *L. casei*-immunized HIL extract against *S. enteritidis*, *S. typhimurium* and *S. pullorum*. Various concentrations of 10 µL melittin (**A**) or 10 µL *L. casei*-immunized HIL extract (**B**) were mixed with 90 µL of liquid media containing 1 × 10^6^ cfu/mL of *S. enteritidis*, *S. typhimurium*, or *S. pullorum* and incubated at 37 °C for 24 h.

**Figure 6 insects-11-00704-f006:**
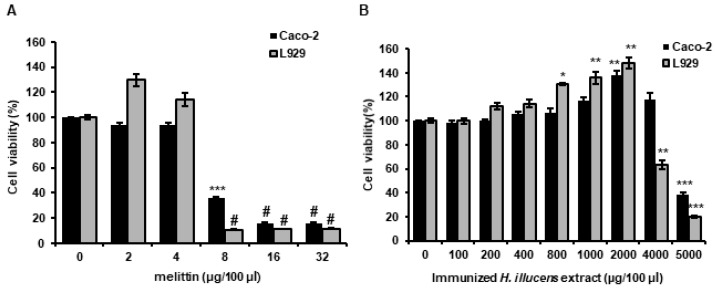
Evaluation of cytotoxic effects of melittin (**A**) or *L. casei*-immunized HIL extract (**B**). (**A**,**B**) 1 × 10^4^ cells were seeded into wells, allowed to attach for 24 h, and then cultured with various concentrations of melittin or *L. casei*-immunized HIL extract for 24 h. The values of column graphs are presented as the mean ± S.D. *, **, *** and # indicate *p* < 0.05, *p* < 0.01, *p* < 0.005, and *p* < 0.001 vs. 0 µg/100 µL, respectively.

**Table 1 insects-11-00704-t001:** Primer sequences used to analyze cecropin 1 and defensin 1 gene expressions by real-time PCR.

Name		Sequences
HiCec1	Forward	5′-TTGGTCAACGAGTTCGTGATGC-3′
	Reverse	5′-TCCTTGTTGTGGTGGTCCACCT-3′
HiDef1	Forward	5′-AGGTGGTGGAGCAGCATTAC-3′
	Reverse	5′-ACGACGTCCCAAAGCAATAC-3′
Act5C	Forward	5′-AAGGACTCGTACGTGGGTG-3′
	Reverse	5′-CATCTTCTCACGGTTGGC-3′

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
