# Peer review of "Antimicrobial Activity of an Extract of Hermetia illucens Larvae Immunized with Lactobacillus casei against Salmonella Species"

_insects, 2020, doi:10.3390/insects11100704_

Round 1
Reviewer 1 Report
The manuscript focus on a very interesting issue related to the AMPs. The immunization of black soldier flies can be an interesting approach to fight against AMR.
The introduction section is well written but it would be better is actual references could also be included (2018-2020).
The M&M section should be improved.
Section 2.1
References to some bacteria are not included. Every strain included in a Culture Collection has a unique identifier code, that is not always in the manuscript ()i.e the five Lactobacillus or the three Salmonella strains)
Section 2.2
The quantity of Lactobacillus inoculated is not included here. These are M&M issues, not result
Section 2.3
Using "by" a microwave. Writing mistake?
Section 2.4.
Coli ---> coli
Salmonella in full only the first time. Salmonella pullorum. The next strains S. is enough
Section 2.5
Melittin and bee venom. Why are they used? At the end of the manuscript it is well described, but maybe some reference in the Introduction would help the readers.
Results
Figure 1 legend
marsescen --> marcescens
Section 3.2. The first phrase should be in M&M rather than here.
Figure 2.
H-illucens --> H. illucens
Section 3.3
I suggest exchanging phrases 1 and 2. We investigated ... and then Salmonella species ...
Which is the base for protease effects? Can you add references to your speculation and explain in-depth this aspect?
Section 3.6
Maybe antimicrobial would be better than antibiotic in the first phrase
Place Figure 5 before section 3.6
Discussion
Man ----> human beings to include not only males (to be polite in gender issues)
Author Response
Thanks for sharpen your indication and comment. Now, we reply for your comments as below.
The manuscript focus on a very interesting issue related to the AMPs. The immunization of black soldier flies can be an interesting approach to fight against AMR.
The introduction section is well written but it would be better is actual references could also be included (2018-2020).
→ Thanks for your comment. We changed and added some references in the introduction.
The M&M section should be improved.
Section 2.1
References to some bacteria are not included. Every strain included in a Culture Collection has a unique identifier code, that is not always in the manuscript ()i.e the five Lactobacillus or the three Salmonella strains)
→ Thanks for your comment. We added the identifier code of each bacterium.
Section 2.2
The quantity of Lactobacillus inoculated is not included here. These are M&M issues, not result
→ Thanks for your comment. We inserted the quantity of Lactobacillus inoculated and removed it in result.
Section 2.3
Using "by" a microwave. Writing mistake? →We used a microwave.
Section 2.4.
Coli ---> coli → Thanks for your comment. We corrected.
Salmonella in full only the first time. Salmonella pullorum. The next strains S. is enough
→ Thanks for your comment. We corrected.
Section 2.5
Melittin and bee venom. Why are they used? At the end of the manuscript it is well described, but maybe some reference in the Introduction would help the readers.
→ Thanks for your comment. We didn’t use bee venom. It was error. We corrected the sentence.
We showed some reference of melitting in the introduction.
Results
Figure 1 legend
marsescen --> marcescens
→ Thanks for your comment. We corrected.
Section 3.2. The first phrase should be in M&M rather than here.
→ Thanks for your comment. We corrected.
Figure 2.
H-illucens --> H. illucens
→ Thanks for your comment. We corrected.
Section 3.3
I suggest exchanging phrases 1 and 2. We investigated ... and then Salmonella species ...
→ Thanks for your comment. We corrected.
Which is the base for protease effects? Can you add references to your speculation and explain in-depth this aspect?
→ Thanks for your comments. We have no reference to supports our speculation. So, we deleted the sentence to prevent because the speculation is not clear.
Section 3.6
Maybe antimicrobial would be better than antibiotic in the first phrase
→ Thanks for your comment. We corrected.
Place Figure 5 before section 3.6
→ Thanks for your comment. We corrected.
Discussion
Man ----> human beings to include not only males (to be polite in gender issues)
→ Thanks for your comment. We corrected.
Reviewer 2 Report
Dear Authors
Manuscript Review OCT 1 2020
Insects MPDI
Article
Antimicrobial activity of an extract of Hermetia illucens larvae immunized with Lactobacillus casei against Salmonella species
Kyu-Shik Lee 1†, Eun-Young Yun 2† and Tae-Won Goo 3*
1 Department of Pharmacology, College of Medicine, Dongguk University, Gyeongju 38766, South Korea; there1@dongguk.ac.kr (K.-S.L.)
2 Department of Integrative Bio-industrial Engineering, Sejong University, Seoul 05006, South Korea; yuney@sejong.ac.kr (E.-Y.Y.)
3 Department of Biochemistry, College of Medicine, Dongguk University, Gyeongju 38766, South Korea; gootw@dongguk.ac.kr (T.-W.G.)
* Correspondence: gootw@dongguk.ac.kr; Tel.: +82-54-703-7801
† These authors contributed equally to this work.
Received: date; Accepted: date; Published: date
Review
I wish to thank the authors for their research.
This manuscript examines the antimicrobial activity of Hermetia illucens larvae extracts that were previous exposed to bacterial strains. The extracts were tested using Radial Diffusion assays and MIC tests. Gene expression was also used to measure certain genes associated with the production of antimicrobial peptides.
Comments:
Materials & Methods
2.2
I think that this was really an infection process versus an immunization process. To what extent are the Lactobacillus or other bacterial strain infected with the bacterial cultures? Are the insects susceptible to infection from these bacterial strains?
Did you measure consequences of infections such as death rate, feeding interference?
How do you know that you stimulated the innate immune response in the insects? Did you use a marker for general innate response?
2.3
The extracts were treated with acid and heat for a time. Is there any enzymatic activity associated with the extracts such as protease or lipase or other metabolic enzymes that survive the process? The enzyme could alter the characteristics of assays.
Based on my interpretation, I do not see a dialysis procedure? So, you are concentrating any and all materials in the extracts including chemicals metabolites. A dialysis procedure would help remove chemicals such as hydrogen peroxide which Lactobacillus species are known to produce. Lactobacillus can also make lactic acids which are antimicrobial.
You could be concentrating hydrogen peroxide which could also have antimicrobial activity. How did you distinguish this from a protein-based activity? You assumed AMP but other chemicals could be present that cause antimicrobial activity in your assays.
Does proteolytic activity interfere with your assays? This would indicate a protein-based product. This was not proven or explored, it was only assumed. Hydrogen peroxide could result in antimicrobial activity in your assays especially if it was concentrated and there was no dialysis step. Hydrogen peroxide is resistant to pH and temperature.
It seems that the authors should spend more time on understanding the nature of this antimicrobial agent(s). Is it protein or chemical? Is it resistant to proteases? I caution the authors in assuming that it is AMP without more evidence. You did not use dialysis top remove chemicals in your extracts. Maybe the AMPs are too small for dialysis and you need other evidence.
2.4 Was the bacterial used in the RDA standardized for each petri plate? How was this accomplished?
2.5
Please explain how you determined the amount of bacterial CFU/ml that was used for the assays?
Results
General
Most of the column graphs and did not specify of the differences seem among values were statistically significant? Please identify which values were statistically significant values in the column graphs. Just indicating that a value was higher often does not relate to statistically significant difference as you know.
It appears from your data i.e. fig 1 and others that the antimicrobial activity was more pronounced with the E. coli which is gram negative versus the S. aureus which is gram positive. Yet your “immunization” used was a gram-positive bacteria. Can you explain this apparent dichotomy?
3.2-3.3 and fig 2 or fig 3
Again, was there a statistical difference among the times in fig 2b?
It seems odd that the gene expression data shown in fig 2c correlates with the data in 2b. Gene expression of mRNA should precede any protein production due to the lag time from mRNA expression and protein synthesis. Usually the protein synthesis should lag of follow after the mRNA synthesis but in this case, fig 2b and fig2c correlate. Can you explain this issue.
Once again, you have not proven that the antimicrobial activity is due to a protein product versus a chemical such a hydrogen peroxide or lactic acid.
What microbe was used in the tests for Fig 3b and 3c? Please make this clear, thank you.
3.4
There seems to be some issue with the presentation of results for figure 4a and 4b.
Fig 4a is only the melittin data and Fig 4b is only the immunized extract data.
There is no data for the fig 4b described in the manuscript for Salmonella species. Please correct this issue.
Fig 4b in the manuscript
Please comment on the fact that for fig 4b in the manuscript, that both bacterial species show the same sensitivity to the antimicrobial substance in spite of the fact that one is gram negative and the other is gram positive. The “immunization” was with gram positive. Usually innate response could distinguish gram positive versus gram negative and have different responses. Please provide a comment on this issue.
Do the AMPs act the same sensitivity with gram positive and gram negative?
3.6
Fig 6a and fig 6b seem to be mixed. Please correct this issue.
Discussion
Please describe your attempt at determining the nature of this antimicrobial substance. Why are you assuming this is a peptide versus a chemical or antimicrobial enzyme such as lysozyme?
Please describe how you determined that the innate immune system was stimulated by your inoculation procedure? What was the innate immune system marker that you used?
Why would the gram-positive bacteria (Lactobacillus) that are beneficial and not cause disease stimulate the insect immune response better than the pathogens that you used? Please provide your thoughts.
The authors should really spend more time on determining the nature of the antimicrobial substance. You should convince readers that this is an AMP versus chemical or another enzyme. The gene expression data is only circumstantial.
Author Response
Thanks for sharpen your indication and comment. Now, we reply for your comments as below.
This manuscript examines the antimicrobial activity of Hermetia illucens larvae extracts that were previous exposed to bacterial strains. The extracts were tested using Radial Diffusion assays and MIC tests. Gene expression was also used to measure certain genes associated with the production of antimicrobial peptides.
Comments:
Materials & Methods
2.2
I think that this was really an infection process versus an immunization process. To what extent are the Lactobacillus or other bacterial strain infected with the bacterial cultures? Are the insects susceptible to infection from these bacterial strains?
Did you measure consequences of infections such as death rate, feeding interference?
How do you know that you stimulated the innate immune response in the insects? Did you use a marker for general innate response?
→ Thanks for your comment. We infected approximately 104 cfu of bacteria with the bacterial cultures. In case of L. casei infection, we did not found death of any H. illucens larvae in 72 h. Therefore, it is not susceptible to H. illucens.
Cecropin 1 and defensin 1 were used as markers for determining the induction of innate immune response because it is well-known that AMPs are dramatically induced when innate immune responses are activated (Ref. Tanji and Ip, Trends in immunology, 26, 2005; Lehrer and Ganz, Current opinion in immunology. 11, 1999; Vilmos and Kurucz, Immunology letters, 62, 1998; Flatt et al., Journal of Experimental Biology, 211, 2008; Tanji et al., Molecular and celluar biology, 27, 2007).
2.3
The extracts were treated with acid and heat for a time. Is there any enzymatic activity associated with the extracts such as protease or lipase or other metabolic enzymes that survive the process? The enzyme could alter the characteristics of assays.
→ Thanks for your comment. The enzymes have no antimicrobial activity and can be fully inactivated by heat and acid during in extraction procedure. Therefore, we think the alteration of AMP activity by the enzymes can negligible.
Based on my interpretation, I do not see a dialysis procedure? So, you are concentrating any and all materials in the extracts including chemicals metabolites. A dialysis procedure would help remove chemicals such as hydrogen peroxide which Lactobacillus species are known to produce. Lactobacillus can also make lactic acids which are antimicrobial.
You could be concentrating hydrogen peroxide which could also have antimicrobial activity. How did you distinguish this from a protein-based activity? You assumed AMP but other chemicals could be present that cause antimicrobial activity in your assays.
→ Thanks for your comments. When we prepare the extract, we boiled for 30 min and dried completely for 9 h in vacuum-spin drier. Therefore, hydrogen peroxide can be fully removed during in the extraction procedure. Furthermore, as shown Fig. 3C, antimicrobial activity of HIL extract was not affected by pH (2~11). This result indirectly supports the antimicrobial activity is not affected by hydrogen peroxide. Lactic acid is another main chemical produced by Lactobacillus species. In this investigation, we infected L. casei into body cavity of H. illucens larvae. Body cavity is not suitable environment to survive and grow the bacterium. Therefore, we think Lactobacillus species cannot survive in the body cavity for a long time. As a result, lactic acid should not be produced fully. Consequently, we thought dialysis procedure were not necessary.
Does proteolytic activity interfere with your assays? This would indicate a protein-based product. This was not proven or explored, it was only assumed. Hydrogen peroxide could result in antimicrobial activity in your assays especially if it was concentrated and there was no dialysis step. Hydrogen peroxide is resistant to pH and temperature.
→ Thanks for your comments. As mentioned above, antimicrobial activity of HIL extract was not affected by pH (2~11). Although, hydrogen peroxide is resistant to pH and temperature, hydrogen peroxide can be removed completely during in the extraction procedure. Most of all, we think hydrogen peroxide and lactic acid should not be produced fully in H. illucens larvae.
It seems that the authors should spend more time on understanding the nature of this antimicrobial agent(s). Is it protein or chemical? Is it resistant to proteases? I caution the authors in assuming that it is AMP without more evidence. You did not use dialysis top remove chemicals in your extracts. Maybe the AMPs are too small for dialysis and you need other evidence.
→ We showed the induction of AMP gene expression by L. casei in H. illucens larvae by real-time PCR and the relationship between AMP gene expression and antimicrobial activity (Figure 2B and C). As shown Figure 1, 2B and 3A, we compare antimicrobial activity of H. illucens larvae extract with that of L. casei-immunized H. illucens larvae extract. We use H. illucens larvae extracts as normal control. The result showed no antimicrobial activity was observed in normal control. It means chemicals do not affect to antimicrobial activity of the extract of L. casei-immunized H. illucens larvae.
2.4 Was the bacterial used in the RDA standardized for each petri plate? How was this accomplished?
→ Thanks for your comment. We determined cfu/ml of each bacterium by tittering with a serial dilution. To determine cfu/ml, we plated serial-diluted each bacterium onto solid medium and grown for appropriate time and at temperature. Then, cfu/ml was calculated.
2.5
Please explain how you determined the amount of bacterial CFU/ml that was used for the assays?
→ We mentioned how to determine cfu/ml of bacteria.
Results
General
Most of the column graphs and did not specify of the differences seem among values were statistically significant? Please identify which values were statistically significant values in the column graphs. Just indicating that a value was higher often does not relate to statistically significant difference as you know.
→ Thanks for your comments. We statistically analyzed data of the column graphs.
It appears from your data i.e. fig 1 and others that the antimicrobial activity was more pronounced with the E. coli which is gram negative versus the S. aureus which is gram positive. Yet your “immunization” used was a gram-positive bacteria. Can you explain this apparent dichotomy?
→ AMPs possesses a net positive charge. Therefore, AMPs can bind to negative-charged component onto bacterial membrane. AMPs are not antibody. It is very well-known AMPs have no specificity. Therefore, the antimicrobial activity against Gram-negative or Gram-positive is not determined by whether the bacteria infected into insects are Gram-negative or Gram-positive.
(Ref. Park et al., Journal of Life Science, 26, 2016; Vieira et al., Parasite & Vectors, 7, 2014)
3.2-3.3 and fig 2 or fig 3
Again, was there a statistical difference among the times in fig 2b?
It seems odd that the gene expression data shown in fig 2c correlates with the data in 2b. Gene expression of mRNA should precede any protein production due to the lag time from mRNA expression and protein synthesis. Usually the protein synthesis should lag of follow after the mRNA synthesis but in this case, fig 2b and fig2c correlate. Can you explain this issue?
→ The lag time from mRNA expression and protein synthesis does not always coincide. This is not an abnormal phenomenon. (Ref.: Ravenna et al., Plos ONE, 2014; Cheng et al., Molecular Systems Biology, 12, 2016; Khodada et al., Behavioural Brain Research, 284, 2015)
Once again, you have not proven that the antimicrobial activity is due to a protein product versus a chemical such a hydrogen peroxide or lactic acid.
→ We commented above about your comment.
What microbe was used in the tests for Fig 3b and 3c? Please make this clear, thank you.
→ Thanks for your comment. We added the information of microbe in figure legend of Fig 3B and 3C.
3.4
There seems to be some issue with the presentation of results for figure 4a and 4b.
Fig 4a is only the melittin data and Fig 4b is only the immunized extract data.
There is no data for the fig 4b described in the manuscript for Salmonella species. Please correct this issue.
→ Thanks for your comment. We made a mistake in Section 3.4. We corrected the section. The result for Salmonella species were explant at Section 3.5.
Fig 4b in the manuscript
Please comment on the fact that for fig 4b in the manuscript, that both bacterial species show the same sensitivity to the antimicrobial substance in spite of the fact that one is gram negative and the other is gram positive. The “immunization” was with gram positive. Usually innate response could distinguish gram positive versus gram negative and have different responses. Please provide a comment on this issue.
→ AMPs do not have specificity to Gram-positive or Gram-negative. AMPs can bind to bacterial membrane and then kill without specificity. It is well-known. When insects are infected by pathogen, the activation of the innate immune responses is mediated by IMD signaling and Toll signaling. In general, IMD signaling is activated by Gran-negative bacterial infection and Gram-positive bacteria activated Toll signaling. However, recent investigations have demonstrated crosstalk between the IMD and Toll signaling in insects.
Do the AMPs act the same sensitivity with gram positive and gram negative?
-> Yes. AMP sensitivity to Gram-positive and Gram-negative is not determined by a type of infected-bacteria.
3.6
Fig 6a and fig 6b seem to be mixed. Please correct this issue.
→ Thanks for your comment. We found the legend was mistyped. We corrected the legend
Discussion
Please describe your attempt at determining the nature of this antimicrobial substance. Why are you assuming this is a peptide versus a chemical or antimicrobial enzyme such as lysozyme?
→ Thanks for your comment. mRNA levels of HiCec1 and HiDef1 were measured to confirm whether innate immune response was activated. The AMPs were used as markers for innate immune response. Lysozyme is also important antimicrobial peptide. However, it is not stable to heat and acid. Therefore, we think the activity of lysozyme was lost during in extraction procedure. Furthermore, we previously tested antimicrobial activity of peptide purified from the extract in against Klebsiella pneumoniae and Shigella dysenteriae. The result showed strong antimicrobial activity to the bacteria. Based on the result, we thought the antimicrobial activity of the extract of L. casei-immunized H. illucens larvae is primarily caused by AMP (Ref. Choi et al., Entomological Research, 48, 2018).
Please describe how you determined that the innate immune system was stimulated by your inoculation procedure? What was the innate immune system marker that you used?
→ Insects lack adaptive immune response. Therefore, innate immune response is only activated by pathogen. AMP induction is a typical humoral response against pathogen and is mediated by both IMD signaling and Toll signaling. So, mRNA of AMP can be used marker for the innate immune response in insects. In our investigation, we also evaluated the change of AMP mRNA expression to confirm whether the innate immune response was induced.
Why would the gram-positive bacteria (Lactobacillus) that are beneficial and not cause disease stimulate the insect immune response better than the pathogens that you used? Please provide your thoughts.
→ Although Lactobacillus species are beneficial, many investigations showed the bacteria stimulate innate immune response in insects. The response by Lactobacillus species contributes to improving survival of insects from pathogen infection. In this investigation, we also found no death of L. case-infected H. illucens. We added the comment in discussion with references. (Furthermore, Lactobacillus species activated innate immune response in insects and contributed to reduction of susceptibility in pathogen infected-insects [26-29].)
The authors should really spend more time on determining the nature of the antimicrobial substance. You should convince readers that this is an AMP versus chemical or another enzyme. The gene expression data is only circumstantial.
→ This investigation is to determine whether antimicrobial activity was enhanced by Lactobacillus species and the extract of immunized H. illucens larvae can be used as antimicrobial substance in livestock industry. This investigation is not to determine AMPs induced by Lactobacillus species.
Round 2
Reviewer 2 Report
Dear Authors
Thank you for your comments to the first manuscript review and your research. I am satisfied with the authors comments.
This manuscript is a resubmission of an earlier submission. The following is a list of the peer review reports and author responses from that submission.